# Financial Planning for Retirement: The Mediating Role of Culture

**Ahmad Ghadwan \*** , **Wan Marhaini Wan Ahmad and Mohamed Hisham Hanifa**

Department of Finance, Faculty of Business and Economics, University of Malaya in Kuala Lumpur, Kuala Lumpur 50603, Malaysia; wmarhaini@um.edu.my (W.M.W.A.); mhisham@um.edu.my (M.H.H.)
* Correspondence: ahsagh1984@gmail.com or aghadwan@taibahu.edu.sa;
  Tel.: +966-535311500 or +60-11-23789041

**Abstract:** The life expectancy rate of individuals worldwide has risen, and Saudi Arabia is not excluded. Rising long-life expectancy may jeopardize employees' pensions and reduce the chances of adequate earnings and a decent life after retirement. Moreover, the number of employees, who have paid into pension funds and are now retired, has increased, indicating that pension funds are expected to decrease. Apart from the above, the level of financial literacy in Saudi Arabia was substandard. Therefore, the ultimate objective of this research is to examine the measurable factors that could impact employees in their financial planning for retirement (FPR). These factors comprise the employee's financial literacy (FL), financial risk tolerance (FRT), and cultural factors based on the CWO model. Moreover, this study aims to investigate the mediating roles of culture in their relationship with financial planning for retirement. Primary data was collected during the COVID-19 pandemic from mid-July 2020 until the end of January 2021 using a non-probability convenience sampling approach involving 525 participants. The Structural Equation Modelling (SEM) technique was used to analyze the data. To determine the type of study variables, either a formative or reflective model of Confirmatory Tetrad Analysis (CTA-PLS) was used. The results show the significant influence of basic FL, FRT, and culture on FPR. Moreover, it shows the critical role of culture among those with advanced FL and FRT. Previous studies have examined FL and FRT in FPR without considering the effect of culture as a mediator.

**Keywords:** financial planning for retirement; financial literacy; financial risk tolerance; culture; intentional change theory; Saudi Arabia

## 1. Introduction

In the last century, vast differences have been seen in the social economies of both developed and developing countries (Jaafar et al. 2019; Liu et al. 2021). These changes have affected the financial well-being of workers (Bacova and Kostovicova 2018; Topa et al. 2018a) and their pre-retirement and post-retirement working lives. Due to these differences, many issues arise in the preparation of the retirement plan as a result of the aging population (e.g., increased life span and a declining fertility rate), financial illiteracy, and so on.

The aging of the population is a challenge for all societies. According to the World Bank, the USA's life expectancy increased from 69 years old in 1960 to almost 79 in 2017, while it increased in Saudi Arabia from around 45 in 1960 to nearly 75 in 2017. In the same vein, The Department of Economic and Social Affairs of the United Nations (2017) indicated that the number of people aged 60 and over is forecast to be 1.4 billion by 2030 and 2.1 billion by 2050. Moreover, Tan and Singaravelloo (2020) indicated that the growth of the population of the elderly has been faster than that of the young due to the present decline in the fertility rate and the increase in life expectancy. In more detail, the median population age in developed and developing economies has risen from 29 to 24 years in 2003 and is

expected to reach 45 and 36 years, respectively, in 2050 (Bongaarts 2004). Therefore, financial planning for retirement (FPR) to have a financially secured post-retirement becomes a grave concern for many developed and emerging countries (Henkens 2022; Scharn et al. 2018; Yeung and Lee 2022). Table 1 below reveals the life expectancy of some developed and developing countries and indicates the higher life expectancy of industrialized countries than emerging countries.

**Table 1.** Life Expectancy at Birth (World Bank 2022).

| | Life Expectancy | | | Life Expectancy | |
|---|---|---|---|---|---|
| **Industrialized Countries** | **1960** | **2019** | **Emerging Countries** | **1960** | **2019** |
| Australia | 70.82 | 82.90 | Argentina | 65.00 | 76.67 |
| Canada | 71.13 | 82.05 | Brazil | 54.14 | 75.88 |
| Germany | 69.31 | 80.94 | China | 43.73 | 76.91 |
| Denmark | 72.18 | 81.20 | Algeria | 46.14 | 76.88 |
| Spain | 69.11 | 83.49 | Mexico | 57.08 | 75.05 |
| United States | 69.77 | 78.79 | Malaysia | 59.99 | 76.16 |
| Italy | 69.12 | 83.20 | Saudi Arabia | 45.64 | 75.13 |

Another challenge that people face is the lack of financial literacy, which is why FPR is essential for their financial security once they leave the workplace. This phenomenon is widespread among employees and retirees in many countries worldwide (Boisclair et al. 2017; Klapper and Lusardi 2020). Among the G20s, according to the OECD (Organization for Economic Co-Operation and Development) (2017), the level of financial literacy in Saudi Arabia was substandard. It also ranked Saudi Arabia last among G20 countries because of poor financial awareness, negative attitudes, and weak ideas for individuals. This finding is shared by different studies that investigated financial literacy among investors (Mian 2014), undergraduate students (Alghamdi et al. 2021; Alyahya 2017), and young adults (Khan and Tayachi 2021) in Saudi. Another study showed that, even though 75% of young people have information on managing their money, only around 11% of adults plan for their expenditures (Al-Ghabri 2013).

In addition to the above-mentioned practical problems facing FPR, there is a lack of studies, especially in Saudi Arabia (Alghamdi et al. 2021; Diaw 2017), designed to investigate the FPR from different perspectives (e.g., finance, psychological, external variables). A critical psychological variable forwarded by literature on people's willingness to adopt an FPR model is traced to the aspect of financial risk tolerance (Kerry and Embretson 2018; Lusardi 2000). These studies have concluded that there was a link between financial risk tolerance, general risk tolerance, and financial and saving practices. Risk tolerance was also a significant predictor of excellent FPR (Larisa et al. 2020), especially in delineating the financial planning activities, such as patterns of investment decisions (Tomar et al. 2021). Despite the critical role financial risk tolerance plays in financial planning, including for retirement purposes, studies focusing on Middle Eastern countries, specifically those in the Gulf States, have not included these aspects. Prior studies have focused mainly on capacity variables, such as financial literacy in the UAE (Alkhawaja and Albaity 2020; Hassan Al-Tamimi and Kalli 2009) and Saudi Arabia (Alghamdi et al. 2021; Diaw 2017; Khan and Tayachi 2021), though none have considered the critical role of financial risk tolerance and culture within financial literacy.

To accomplish this, a review of the empirical works on FPR worldwide sheds light on a few important aspects regarding FPR. Recent studies have concluded that the CWO model is one of the best models to assess the FPR by examining capacity, willingness, and opportunity variables for retirement saving behaviors (Topa et al. 2018a; Ghadwan et al. 2022). It enables researchers to examine FPR behavior, especially the interaction between several variables. Based on the above, the present study aims to develop the methodology of financial risk perception regarding FPR. Moreover, it expands the retirement planning

literature by investigating financial literacy, financial risk tolerance, and the culture of the employees' financial planning and the roles these variables play in the retirement behavior of employees in all of Saudi Arabia's public universities by using the CWO model (Hershey et al. 2012).

This paper is divided into nine sections. Section 2 discusses the CWO model. Section 3 presents the theoretical background and the hypotheses' development. Section 4 outlines the research design, instruments, participants, and analysis procedure. Section 5 highlights the key findings, while Section 6 explains the study's results and interpretations. Sections 7 and 8 provide the implications, recommendations, and limitations, whereas Section 9 provides the conclusion.

## 2. The CWO Model

Hershey et al. (2012) developed a modified version of the model known as "Capacity-Willingness-Opportunity Model" (CWO) for job performance, proposed by Blumberg and Pringle (1982) for the specific review of FPR research. The CWO model is structured on three dimensions, i.e., capacity, willingness, and opportunity, to capture sets of factors that help workers invest, plan, and save successfully for retirement. The dimension of capacity includes perceptive variables and skills that assist in distinguishing people's abilities in their knowledge and skills required to save, invest, and plan for retirement (Hershey et al. 2012; Topa et al. 2018a). The dimension of willingness comprises psychological and emotional variables which can motivate people to plan and save for retirement. Lastly, the dimension of opportunity includes external influence variables, such as parental influence. In the current study, financial literacy represents the capacity dimension, financial risk tolerance represents the willingness dimension, while culture is considered as the opportunity dimension in the CWO model.

Topa et al. (2018a) recommended that the model be used for comprehending employees' FPR behaviors for several reasons. First, it is specifically intended to be used for FPR interpretation. Second, the dimensions presented by the model make it possible to understand in detail the behaviors of retired people in their FPR by integrating extra factors to grasp the motivating effects of the determinants (Topa et al. 2018a). Furthermore, it is appropriate to examine various economies with various cultural and political environments due to the ability of the model to change with the lives of individuals who change over time, representing the continuity of predispositions for change throughout adulthood (Hershey et al. 2012). Finally, most importantly, the CWO model surpasses the weaknesses of other FPR models (Ghadwan et al. 2022). Simultaneously, it evaluates how factors in various fields are related to FPR.

## 3. Theoretical Background and Hypotheses Development

### 3.1. Intentional Change Theory (ICT)

The theoretical foundation for the present study is drawn from the Intentional Change Theory (ICT), which was developed by Richard Boyatzis (2006). Boyatzis (2019) and Topa et al. (2018a) proposed that combining ICT with the variables of capacity, willingness, and opportunity model (Hershey et al. 2012) would increase the understanding of the process, antecedents, and outcomes of FPR. The capacity dimension, which includes the cognitive variables, can predict a sustainable change achieved through financial knowledge and skills, and this is a prerequisite to an individual's readiness for FPR (Topa et al. 2018a). For example, when an individual intends to plan for retirement, the individual needs to have some financial knowledge and skills so as to achieve several necessary financial decisions before retirement. Studies such as Lusardi (2019) and Rudi et al. (2020) concluded that life's financial decisions during the college-to-career transition and after retirement rely on the individuals' financial literacy and their comprehension of personal finance. Lusardi (2019) stated that life's financial decisions rely on the individuals' financial knowledge and skills and the comprehension of personal finance.

### 3.2. Financial Literacy (FL)

One of the most crucial factors in retirement planning is financial literacy (Gallego-Losada et al. 2022; Klapper and Lusardi 2020). Financial literacy and retirement planning have been closely examined in developed countries, such as in the United States of America (Lusardi 2008; Lusardi and Mitchell 2011a), Canada (Boisclair et al. 2017), and Poland (Swiecka et al. 2020), as well as in developing countries, such as Saudi Arabia (Alyahya 2017; Khan and Tayachi 2021), Malaysia (Selamat et al. 2020; Tan and Singaravelloo 2020), along with Brunei (Salleh and Baha 2020). These studies, along with others, have highlighted the value of financial literacy in finance and other fields.

In terms of retirement, those who were financially literate were expected to better plan for retirement, since they were more knowledgeable about the power of interest compounding and were able to make calculations (Hutabarat and Wijaya 2020; Van Rooij et al. 2012; Topa et al. 2018b). Such an ability helps an individual to have different financial resources with which to generate funds besides social security and pension to increase their economic well-being in the future (Palací et al. 2018; Rudzinska-Wojciechowska 2017). Mastering these skills and knowledge allows individuals to effectively understand and manage financial assets, which must be reflected in their well-being. Sarigul (2014) proved that understanding financial literacy helps individuals to manage revenues and expenditures using different financial instruments and tools to increase their wealth and financial security. As a result, it is more important than ever to determine individuals' level of financial literacy and how it affects retirement planning (Lusardi 2019).

Based on Lusardi (2008), basic financial literacy is the minimum level of personal financial planning that everyone needs to know about, such as the time value of money, inflation, numeracy, the money illusion, and compound interest. According to the literature, some studies showed that basic financial literacy is adequate and has a significant positive relationship with FPR (Boisclair et al. 2017; Lusardi and Mitchell 2008; Ricci and Caratelli 2017). Meanwhile, advanced financial literacy involves several topics, such as stocks and bonds, financial institutes, mutual funds, interest rate effect on securities, and risk–return relationship issues.

Others, however, found that only advanced financial literacy had a significant positive relationship with FPR (Almenberg and Save-Soderbergh 2011; Baker et al. 2020; Brahmana et al. 2016; Van Rooij et al. 2011a), while others (Baker et al. 2020; Brahmana et al. 2016; Crossan et al. 2011; Van Rooij et al. 2011a) indicated no significant relationship between basic financial literacy and retirement planning. These results indicate that the financial literacy literature did not provide consistent results. Based on the literature which finds a positive relationship between financial literacy and FPR above, the following assumptions are made for this review:

**H₁.** *Basic financial literacy has a significant positive relationship with financial planning for retirement.*

**H₂.** *Advanced financial literacy has a significant positive relationship with financial planning for retirement.*

### 3.3. Financial Risk Tolerance (FRT)

Financial risk tolerance is the most crucial variable for those planning their financial operations and is, at the same time, the most challenging variable to evaluate (Cooper et al. 2014). A recent study found that financial risk tolerance is essential for retirement planning and financial counseling (Bayar et al. 2020). Grable (1999, 2000) found that people's financial risk tolerance levels differed based on age, educational background, marital status, occupation, cultural background, and economic expectations. Hence, the determination of personal financial risk tolerance demonstrates the degree to which individuals are able to choose financial investments within their portfolios (Bayar et al. 2020).

The literature indicates that the studies have demonstrated a positive and a negative relationship between financial risk tolerance and FPR. Jacobs-Lawson and Hershey (2005)

and Larson et al. (2016) illustrated that higher levels of financial risk tolerance are positively significant with savings behavior, while Tomar et al. (2021) concluded that the relationship between them was negative. Meanwhile, Croy et al. (2010), Koposko et al. (2015), Hershey et al. (2017), Alkhawaja and Albaity (2020), and Larisa et al. (2020) stated that the relationship between risk tolerance and saving practices was not significant. Based on previous studies on financial risk tolerance and how it relates to retirement planning, the following assumption is proposed:

**H3.** *Financial risk tolerance has a significant positive relationship with financial planning for retirement.*

### 3.4. Mediating Role of Culture

The planning process is different from one country to another, not only because of the country's rules and pension system, but also because of the people's culture. Thus, it is considered complicated. Previous cross-cultural studies looking at retirement had derived some interesting findings. They illustrated that culture-specificities could lead to different preparations regarding retirement planning among the people (Koposko et al. 2016; Weisfeld-Spolter et al. 2018), even with well-developed pension schemes (Lusardi and Mitchell 2011b). For instance, Koposko et al. (2016) compared subjects in the United States and Mexico to see how they behaved in their FPR. Their findings have shown that future time perspective and parental influence on savings were higher among American students than Mexican students. However, retirement goal clarity was higher for Mexican students than US students. Another study investigated the differences in employees' retirement attitudes between the United States and The Netherlands (Hershey et al. 2007). The results have indicated that people in The Netherlands had lower levels of retirement goal clarity and retirement-planning activities. However, their perception of retirement savings was higher when compared to people in the United States.

A study by AXA (2007) conducted its annual report on retirement in several countries worldwide to examine workers' attitudes toward retirement and obtain data on retirees' retirement. The results revealed that retirement varies from country to country because of each society's different customs and traditions. In terms of cultural approaches to retirement, for example, working people in Britain start planning for retirement at 28, Chinese at about 37, and Europeans at about 32 years old. The monthly retirement savings plans in France were 13%, in Australia 37%, and in Belgium 25%. Regarding social approaches to retirement, the report indicated that working individuals in Spain and Germany anticipate retiring at 63, Americans at about 64, and Chinese at about 55 years old. Moreover, it indicated that 34%, 80%, 68%, and 65% of retirees retired before the legal minimum age in France, Italy, Canada, America, and Australia, respectively.

Likewise, Imamoglu et al. (1993) examined aging and retirement attitudes between Sweden and Turkey. The findings revealed that Turkish people became more sociable as they came close to retirement compared to the Swedish people. However, Turkish individuals were less life satisfied in retirement when compared to their Swedish counterparts. Such studies implied that FPR behaviors among various cultures and nations need further investigation in order to establish a more precise picture (Koposko et al. 2016). Thus, the following hypotheses were proposed:

**H4a.** *Culture variable has a significant positive relationship with financial planning for retirement.*

**H4b.** *Culture mediates the relationship between basic financial literacy and financial planning for retirement.*

**H4c.** *Culture mediates the relationship between advanced financial literacy and financial planning for retirement.*

**H4d.** *Culture mediates the relationship between financial risk tolerance and financial planning for retirement.*

## 4. Methodology

### 4.1. Research and Sampling Design

The study's research design is cross-sectional to provide a clear "glance" at the characteristics of the study and associated variables at a single point in time. The quantitative research methodology was used to collect data for the study following the convenience sampling technique due to some justifications. First, although empirical research of superannuation has been criticized for applying convenience samples impacted by unknown selectivity (Topa and Valero 2017), it is commonly applied in personal financial planning (França and Hershey 2018; Safari et al. 2016; Shreevastava and Brahmbhatt 2020). Second, one objective of the present research is to apply the previous research indicators. Highhouse and Gillespie (2010) recommended applying the convenience sampling technique in the case of theory or scale examination.

Third, such a sampling technique allows researchers access to a significant number of individuals in a limited time (Tharenou et al. 2007). The sampling frame only includes the university website for public university employees, leading to a lack of access to information about them in government universities. It is challenging to classify public university employees as academics and non-academics because the universities are different from one city to another. Thus, it was not easy to have a list of all of them to choose names on a random basis. Given this additional constraint, the study followed convenience sampling, as it appeared to be the only practical way of selecting respondents from the sampling frame.

Saudi Arabia has a total of 29 public universities. The present study has examined all public university employees for the following reasons. First, public university employees comprise 48% of government employees and are considered to be the government's top qualified workers. Moreover, the potential for knowledge sharing within universities exceeds that of other sectors (Chahal and Savita 2014), anticipating easy access to knowledge related to personal financial planning regarding retirement. Third, they draw interest in research related to personal financial planning, particularly retirement, because they are registered with Saudi's Public Pension Agency (PPA), mandating pensions after retirement. Fourth, there are no complications, and no formal letters are required to engage the public university employees in this research compared to the military or the banking sector, providing an unbiased response from them.

Due to the COVID-19 pandemic, invitations were sent out to public university employees by distributing an online questionnaire to their formal email addresses. The survey questionnaire was initially distributed between 15 March and 15 June 2020 to all 29 public universities in Saudi Arabia. However, the collection period was extended until late January 2021 because the response rate was insufficient for adequate statistical analysis. A total of 558 individuals responded as participants in this study. The samples were considered sufficient because the number exceeded 382, as indicated by Krejcie and Morgan's (1970) table. Following the removal of outliers, the final study sample was 525, representing a response rate of 43%.

The Intentional Change Theory (ICT) was applied as a theoretical foundation. It was then combined with the CWO model, which focuses on the individuals when explaining an individual's intention to change their behavior and how individuals improve their financial knowledge during the work period before retirement.

### 4.2. Instruments

The study's instruments were modified from earlier studies. For instance, the concept of financial literacy was measured using objective questions adopted from Van Rooij et al. (2011b). Such questions were coded as 1 for the correct answer and 0 otherwise. The first five basic financial literacy scales were applied to capture people's ability to cope with basic financial literacy concepts, including simple calculation, compound interest rate, inflation rate, time value of money, and money illusion. The second five advanced financial literacy scales were applied to capture the individuals' sophisticated financial literacy concepts

in financial assets such as stock and bonds, the trade-off between risk and return, risk diversification, and the function of the stock market.

As the scale of financial risk tolerance has evolved, the development of a 7-item scale (1 = "strongly disagree", 7 = "strongly agree") was chosen based on five items of risk evaluation, as proposed by Jacobs-Lawson and Hershey (2005). Due to the Cronbach alpha level of the scale being 0.83, this scale was thus adopted to measure the ability of individuals to tolerate financial risk.

As for culture, a 16-item scale was adopted from Hofstede's study (Sharma 2010) using two dimensions: uncertainty avoidance and long-term orientation. Uncertainty avoidance is represented by risk aversion and ambiguity intolerance as subdimensions, while tradition and prudence subdimensions represent the long-term orientation dimension. Each subdimension is represented by four items, with one composite score index representing each subdimension.

A composite score is a technique that measures variables with a collection of items into one single concept that acts as proxy variables that are more adequate for representing all of the various aspects of the concept, thereby reducing the measurement error. This approach was applied because it is more precise, as Hair et al. (2017a) have recommended. The questionnaire structure and questions are presented in Table 2.

**Table 2.** Summary of the Measurement Variables.

| |
| --- |
| **Financial Planning for Retirement Questions with Likert Scale (1 = "strongly Disagree", 7 = "Strongly Agree") Cronbach's Alpha = 0.86** |
| 1. I have put aside some money for my retirement.<br>2. I am expecting benefits that can be utilized for my retirement planning.<br>3. I will receive fixed payments as my pensions when I retire.<br>4. I will have enough money to maintain my desired standard of living when I retire.<br>5. I am expecting that I will have enough savings to pay for my expenditures during my retirement.<br>6. I am expecting some earnings, which I can utilize during my retirement. |
| **Financial Literacy coded as 1 for the correct answer and 0 for the incorrect answer** |
| 1. Suppose you had SR 100 in an investing account, and the profit rate was 2% per year. After five years, how much do you think you would have in the account if you left the money to grow?<br>2. Imagine that the profit rate on your investing account was 1% per year and inflation was 2% per year. After one year, how much would you be able to buy with the money in this account?<br>3. Suppose you had SR 100 in an investing account and the profit rate is 20% per year, and you never withdraw money or profit payments. After five years, how much would you have on this account in total?<br>4. Suppose that in the year 2017, your income has doubled, and the prices of all goods have doubled too. In 2017, how much will you be able to buy with your income?<br>5. Assume my friend inherits SR 10,000 today, and his sibling inherits SR 10,000 3 years from now. Who is richer because of the inheritance?<br>6. What happens if somebody buys the stock from Firm B in the stock market?<br>7. What happens if somebody buys a bond of Firm B?<br>8. Considering a long time period (for example, 10 or 20 years), which asset typically gives the highest return?<br>9. When an investor spreads his/her money among different assets, the risk shall __________.<br>10. Which of the following statements describes the main function of the stock market? |
| **Financial Risk Tolerance Questions with a Likert scale (1 = "strongly disagree", 7 = "strongly agree") Cronbach's alpha is 0.87** |
| 1. I am willing to risk financial losses.<br>2. I prefer investments that have higher returns, even though they are riskier.<br>3. The overall growth potential of retirement investment is more important than the level of risk of the investment.<br>4. I am very willing to make risky investments to ensure financial stability in retirement.<br>5. I would never choose the safest investment when planning for retirement. |

**Table 2.** *Cont.*

| Cultural items with a Likert scale (1 = "strongly disagree", 7 = "strongly agree") Cronbach's alpha is 0.79 |
| --- |
| 1. I tend to avoid talking to strangers, especially about my financial matter. |
| 2. I prefer a routine way of life to an unpredictable one full of change. |
| 3. I would not describe myself as a risk-taker. |
| 4. I do not like taking too many chances to avoid making a mistake. |
| 5. I find it difficult to function without clear directions and instructions. |
| 6. I prefer specific instructions to broad guidelines. |
| 7. I tend to get anxious quickly when I do not know the outcome. |
| 8. I feel stressed when I cannot predict the consequences. |
| 9. I am proud of my culture. |
| 10. Respect for tradition is important to me. |
| 11. I value a strong link to my past. |
| 12. Traditional values are important to me. |
| 13. I believe in planning for the long-term. |
| 14. I will work hard for success in the future. |
| 15. I am willing to give up today's fun for success in the future. |
| 16. I do not give up easily, even if I do not succeed in my first attempt. |

### 4.3. Participants

This study focuses on government employees attached to the ministry of higher education in Saudi Arabia. This sector was selected for several reasons. First, they represent 48% of the most significant proportion of government employees. Second, they are more flexible than other sectors, such as the military or banking. By way of clarification, there are no complications, and no formal letters are required to engage the faculty and administrators in this study. Another reason for choosing university employees is because the faculty appears to be more educated and skillful.

### 4.4. Procedure

In this study, Structural Equation Modeling (SEM) has been applied to explain the variance among independent variables and the research hypothesis (Hair et al. 2017a). It could also uncover the variables which contributed to FPR practices. The PLS-SEM has the advantage of conducting Confirmatory Tetrad Analysis CTA-PLS, which can determine the measurement model (Hair et al. 2017a) that can confirm whether the study's conceptual model is formative or reflective. It is also considered to be the most appropriate for multigroup analysis. Conducting the SEM-PLS requires a complete measurement model and structural model. Once the measurement model (outer model) has been evaluated, the second step is to analyze the structural model.

## 5. Results and Findings

### 5.1. Preliminary Data Analysis

Preliminary analysis was conducted, and the goodness of data was examined by screening, cleaning, and coding the data. It helps to ensure that specific assumptions of inferential statistics can be fulfilled. The current study has tested all the variables by applying the standardized (z) score technique to identify the outliers totaling 33. After deleting 33 cases from the original gathered study's data, the data size was reduced to 525. Next, normality was tested using the Mardia (1970) test to measure multivariate skewness and kurtosis. The result assists in verifying whether the study variables were abnormally distributed or not. The results have indicated that Mardia's multivariate skewness and kurtosis had a *p*-value < 0.05, which means that the multivariate study's data were not normally distributed. This result suggests a strong justification for the study to apply PLS-SEM rather than Amos or SPSS. In terms of common method bias (CMB), Harman's Single-Factor Test technique was also conducted. The results have found that the total variance explained by the first single factor was 12.135%. It was far below the curve of the

criteria for common method bias, which was 50%. Based on this, it can be said that the dataset for this study was not suffering from CMB threats.

### 5.2. Descriptive Statistic Result

The majority of respondents were from academia; hence, they are familiar with the google drive platform. The respondents comprised 525 in total, 429 (82%) were married, and more than half (54%) were between 31 and 40 years old. Of this number, 344 (66%) were males. This more or less captured the proportion of the workforce in the government sector, where more than half (59.37%) of the 1,226,700 employed were men (SAMA 2019). In terms of their educational level, most of the participants possessed a postgraduate degree, with 152 (29%) having doctoral degrees and 185 (35%) having master's degrees. Only 161 (31%) were bachelor's degree holders. Table 3 below shows the full results of the demographic profiles of the respondents.

**Table 3.** Respondents' Demographic Profile.

| Demographic Variables | Valid Percent |
| --- | --- |
| **Gender** | |
| Male | 344 (66%) |
| Female | 181 (34%) |
| **Marital Status** | |
| Single | 73 (14%) |
| Married | 429 (82%) |
| Divorced | 23 (4%) |
| **Age** | |
| 26–30 | 74 (14%) |
| 31–35 | 151 (29%) |
| 36–40 | 129 (25%) |
| 41–45 | 60 (11%) |
| 46–50 | 55 (10%) |
| 51–55 | 32 (6%) |
| 56–60 | 16 (3%) |
| Above 60 | 8 (2%) |
| **Employment Sector** | |
| Academic | 330 (63%) |
| Administrator | 195 (37%) |
| **Education** | |
| Secondary | 12 (2%) |
| Diploma | 15 (3%) |
| Bachelor | 161 (31%) |
| Master | 185 (35%) |
| Doctoral | 152 (29%) |

Source: Author Calculations.

### 5.3. Confirmatory Tetrad Analysis (CTA)

The CTA is a technique based on an appraisal of construct items in the form of the tetrad. Tetrads are reflective only if all the tetrads values are non-significantly different from zero; otherwise, these tetrads are to be modeled formatively (Garson 2016; Hair et al. 2017a). In other words, if the correlation between indicators is high and exchangeable, it is considered a reflective measurement model. Otherwise, indicators are considered a formative measurement model (Hair et al. 2017a; Ramayah et al. 2018). Theoretical considerations must verify any alteration of the measurement model.

Table 4 illustrates the CTA-PLS outcomes (5000 bootstrap subsamples). It confirms that some tetrads for each variable were significant, thereby providing empirical support that this study's conceptual model was formative.

**Table 4.** Confirmatory Tetrad Analysis.

| Variable | Tetrad | Original Sample | T Statistics | CI Low Adj. | CI Up Adj. |
|---|---|---|---|---|---|
| Financial Planning for Retirement (FPR) | FPR1, FPR2, FPR3, FPR4 | 0.022 | 0.061 | −0.893 | 0.943 |
| | FPR1, FPR2, FPR4, FPR3 | 0.052 | 0.139 | −0.908 | 1.014 |
| | FPR1, FPR2, FPR3, FPR5 | −0.128 | 0.395 | −0.945 | 0.701 |
| | FPR1, FPR3, FPR5, FPR2 | −0.247 | 0.671 | −1.187 | 0.682 |
| | FPR1, FPR2, FPR3, FPR6 | −0.323 | 1.080 | −1.082 | 0.437 |
| | FPR1, FPR2, FPR4, FPR5 | 1.613 | 3.917 | 0.578 | 2.671 |
| | FPR1, FPR2, FPR5, FPR6 | 0.803 | 2.082 | −0.172 | 1.787 |
| | FPR1, FPR3, FPR4, FPR6 | 1.033 | 2.837 | 0.114 | 1.963 |
| | FPR1, FPR3, FPR6, FPR5 | 1.259 | 3.167 | 0.252 | 2.272 |
| Basic Financial Literacy (BFL) | FL1, FL2, FL3, FL4 | −0.002 | 2.442 | −0.005 | −0.000 |
| | FL1, FL2, FL4, FL3 | 0.000 | 0.466 | −0.001 | 0.002 |
| | FL1, FL2, FL3, FL5 | −0.001 | 1.646 | −0.003 | 0.001 |
| | FL1, FL3, FL5, FL2 | 0.001 | 1.498 | −0.001 | 0.003 |
| | FL1, FL3, FL4, FL5 | 0.001 | 1.592 | −0.001 | 0.004 |
| Advanced Financial Literacy (AFL) | FL8, FL9, FL10, FL6 | 0.000 | 0.083 | −0.002 | 0.002 |
| | FL8, FL9, FL6, FL10 | 0.000 | 0.051 | −0.002 | 0.002 |
| | FL8, FL9, FL10, FL7 | 0.000 | 0.312 | −0.002 | 0.002 |
| | FL8, FL10, FL7, FL9 | −0.001 | 0.780 | −0.003 | −0.001 |
| | FL8, FL10, FL6, FL7 | 0.001 | 1.489 | −0.001 | 0.003 |
| Financial Risk Tolerance (FRT) | FRT1, FRT2, FRT3, FRT4 | 0.510 | 2.097 | −0.041 | 1.094 |
| | FRT1, FRT2, FRT4, FRT3 | 0.451 | 1.871 | −0.109 | 1.017 |
| | FRT1, FRT2, FRT3, FRT5 | 0.694 | 3.133 | 0.183 | 1.218 |
| | FRT1, FRT3, FRT5, FRT2 | −0.156 | 0.889 | −0.568 | 0.249 |
| | FRT1, FRT3, FRT4, FRT5 | −0.244 | 1.194 | −0.726 | 0.228 |
| Culture | Ambiguity Intolerance, Prudence, Risk Aversion, Tradition | −0.129 | 4.196 | −0.188 | −0.071 |
| | Ambiguity Intolerance, Prudence, Tradition, Risk Aversion | −0.020 | 1.026 | −0.059 | 0.018 |

Source: Author Calculations.

### 5.4. Assessing Formative Measurement Model

Hair et al. (2017a) indicated that there was more than one step to examine the formative measurement model, i.e., by examining the convergent validity (redundancy analysis), the collinearity issues (VIF), and, finally, by investigating the significance and relevance of

the formative items (bootstrap). The questionnaire items used to measure the formative variables mentioned above examined the different components of these latent constructs in this study. For example, the time value of money and money illusion are two different measurements of basic financial literacy. Hence, the correlation level among these indicators is not supposed to be high. Therefore, traditional reliability and validity assessments (redundancy analysis) would not necessarily be used in evaluating convergent validity. Instead, formative items were only evaluated for multicollinearity (VIF) and the indicators' weight significance.

    The correlation between the reflective indicators was desired but not for the formative indicators since high correlations between formative indicators influenced items' weights and statistical significance (Hair et al. 2017a; Ramayah et al. 2018). Given that collinearity did not reach the critical level among the formative indicators, calculating the next step in the path model using PLS was no issue. Table 5 presents the formative measurement values, including the VIF. For the third step, the significance and relevance of the outer weights of the formative structures were essential for testing 5000 bootstrap sub-samples. The outcomes have revealed that all the formative indicators were significant at level 5%. However, the outer weights or loading for basic financial literacy ($FL_2$, $FL_3$, $FL_5$) and advanced financial literacy ($FL_7$) were non-significant. It was found that they did not achieve the formative measurement criteria.

**Table 5.** Evaluation of Formative Measurement Model.

| Construct | Items | Outer Weights | T-Value | Outer Loading | T-Value | VIF | Result | Decision |
|---|---|---|---|---|---|---|---|---|
| Financial Planning for Retirement (FPR) | FPR1 | 0.323 | 3.126 | 0.671 | 8.329 | 1.259 | Significant | Kept |
| | FPR2 | 0.071 | 0.642 | 0.534 | 5.902 | 1.320 | Significant | Kept |
| | FPR3 | 0.246 | 2.220 | 0.617 | 7.451 | 1.283 | Significant | Kept |
| | FPR4 | 0.283 | 2.530 | 0.766 | 12.605 | 1.927 | Significant | Kept |
| | FPR5 | 0.245 | 1.907 | 0.779 | 12.398 | 2.107 | Significant | Kept |
| | FPR6 | 0.262 | 2.323 | 0.707 | 9.266 | 1.492 | Significant | Kept |
| Basic FL (BFL) | $FL_1$ | 0.388 | 1.646 | 0.486 | 2.208 | 1.149 | Significant | Kept |
| | $FL_2$ | −0.306 | 1.182 | 0.002 | 0.009 | 1.122 | Non-Significant | Kept |
| | $FL_3$ | −0.346 | 1.476 | −0.105 | 0.448 | 1.091 | Non-Significant | Kept |
| | $FL_4$ | 0.859 | 4.571 | 0.837 | 5.379 | 1.179 | Significant | Kept |
| | $FL_5$ | 0.159 | 0.689 | 0.355 | 1.639 | 1.137 | Non-Significant | Kept |
| Advanced FL (AFL) | $FL_6$ | 0.403 | 1.350 | 0.632 | 2.476 | 1.254 | Significant | Kept |
| | $FL_7$ | −0.182 | 0.567 | 0.327 | 1.325 | 1.312 | Non-Significant | Kept |
| | $FL_8$ | 0.376 | 1.275 | 0.553 | 2.052 | 1.084 | Significant | Kept |
| | $FL_9$ | 0.331 | 1.114 | 0.592 | 2.284 | 1.162 | Significant | Kept |
| | $FL_{10}$ | 0.527 | 1.698 | 0.762 | 2.909 | 1.227 | Significant | Kept |
| Financial Risk Tolerance (FRT) | $FRT_1$ | 0.289 | 2.080 | 0.726 | 8.258 | 1.536 | Significant | Kept |
| | $FRT_2$ | 0.297 | 2.008 | 0.766 | 9.931 | 1.757 | Significant | Kept |
| | $FRT_3$ | 0.274 | 2.305 | 0.688 | 8.283 | 1.398 | Significant | Kept |
| | $FRT_4$ | 0.011 | 0.072 | 0.682 | 7.433 | 1.861 | Significant | Kept |
| | $FRT_5$ | 0.448 | 3.547 | 0.818 | 12.098 | 1.534 | Significant | Kept |
| Culture | Ambiguity Intolerance | 0.238 | 2.694 | 0.717 | 12.104 | 1.496 | Significant | Kept |
| | Prudence | 0.346 | 3.774 | 0.812 | 18.052 | 1.736 | Significant | Kept |
| | Risk Aversion | 0.343 | 4.031 | 0.746 | 13.655 | 1.427 | Significant | Kept |
| | Tradition | 0.360 | 3.599 | 0.814 | 16.791 | 1.738 | Significant | Kept |

Source: Author Calculations.

    However, past literature (Van Rooij et al. 2012; Warmath and Zimmerman 2019) has provided strong evidence for the basic and advanced financial literacy measurements which capture behavior change in general behavior. This seems to be specific to FPR. Based on

the above, the indicators that did not accomplish formative measurement requirements (e.g., outer weight, outer loading) were retained.

### 5.5. Assessment of Structural Model

To evaluate the structural model, five phases were followed: (1) collinearity evaluation (VIF), (2) significance of the path coefficients (*p*-value), (3) the level of the coefficient of determination ($R^2$), (4) effect size ($f^2$), and (5) predictive relevance $Q^2$, as recommended (Hair et al. 2019; Ramayah et al. 2018).

Table 6 shows that the largest VIF value was 1.634, which indicated that, as long as the value of VIF was less than 3, the multicollinearity between the variables was not a concern. The study's hypotheses were examined by using the bootstrap 5000 sub-sample technique. Table 6 and Figure 1 show the path coefficient results. The SEM results confirmed the hypothesized relationship between basic FL, FRT, culture, and FPR. It also showed a significant positive relationship between basic FL ($\beta$ = 0.121, *p* < 0.01), FRT ($\beta$ = 0.122, *p* < 0.01), culture ($\beta$ = 0.356, *p* < 0.01), and FPR. Therefore, $H_1$, $H_3$, and $H_{4a}$ were supported. However, the second hypothesis was rejected because it showed a negative and a non-significant relationship between advanced financial literacy and FPR ($\beta$ = −0.002, *p* > 0.05). Consequently, advanced financial literacy was noted as a non-predictor of FPR; it did not help respondents plan and save for their future.

**Table 6.** Evaluation of Structural Model.

| Hypothesis | Relationship | Std Beta | Std. Dev. | t-Value | *p*-Value | VIF | Decision | F$^2$ |
|:---:|:---:|:---:|:---:|:---:|:---:|:---:|:---:|:---:|
| $H_1$ | Basic FL -> FPR | 0.116 | 0.063 | 1.843 | 0.033 | 1.025 | Supported | 0.017 |
| $H_2$ | Advanced FL -> FPR | −0.002 | 0.054 | 0.046 | 0.482 | 1.034 | Non-Supported | 0.000 |
| $H_3$ | FRT -> FPR | 0.122 | 0.050 | 2.412 | 0.008 | 1.170 | Supported | 0.132 |
| $H_{4a}$ | Culture -> FPR | 0.356 | 0.047 | 7.645 | 0.000 | 1.205 | Supported | 0.016 |
| $H_{4b}$ | Basic FL -> Culture -> FPR | 0.045 | 0.028 | 1.601 | 0.055 | - | Non-Supported | - |
| $H_{4c}$ | Advance FL -> Culture -> FPR | 0.039 | 0.020 | 1.922 | 0.028 | - | Supported | - |
| $H_{4d}$ | FRT -> Culture -> FPR | 0.128 | 0.023 | 5.507 | 0.000 | - | Supported | - |

Source: Author Calculations.

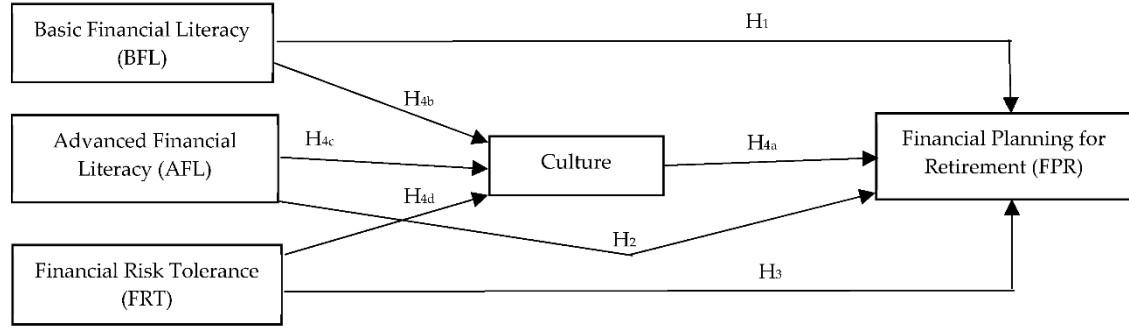

**Figure 1.** Proposed Model of the Study.

With regards to the mediation role of culture, Table 6 shows that culture only mediated the path between advanced FL ($\beta$ = 0.039, *p* < 0.05), FRT ($\beta$ = 0.128, *p* < 0.01), and FPR. Hence, only $H_{4b}$ was not supported. This means that cultural values explain the relationship between those who are knowledgeable in personal finance and those who have the capacity to withstand financial risks when it comes to FPR practices.

To compute the coefficient of the value of the determinant ($R^2$), a smart-PLS algorithm was used. Figure 2 displays the $R^2$ value as 0.201, which was considered to be a moderate value based on Cohen (1988). It further implies that this value had accomplished adequate explanatory power. Assessing the value of $R^2$ for the FPR revealed that the independent

variables, namely financial literacy, financial risk tolerance, and culture, accounted for 20.1% of the variance in the financial planning for the retirement variable. By looking at the $f^2$ values in Table 6, it was noted that FRT had a small effect ($f^2 = 0.132$), and that basic financial literacy ($f^2 = 0.017$) and culture ($f^2 = 0.016$) had a trivial effect, while advanced FL ($f^2 = 0.000$) had no effect in producing $R^2$ for FPR behavior based on Cohen (1988). Figure 2 and Table 6 summarize the results of the structural model. According to Hair et al. (2017b), the blindfolding process must be employed only on an endogenous variable with a reflective indicator. Based on the Confirmatory Tetrad Analysis (CTA), the measurements of FPR were, thus, formative. Therefore, predictive relevance $Q^2$ for this study was not conducted.

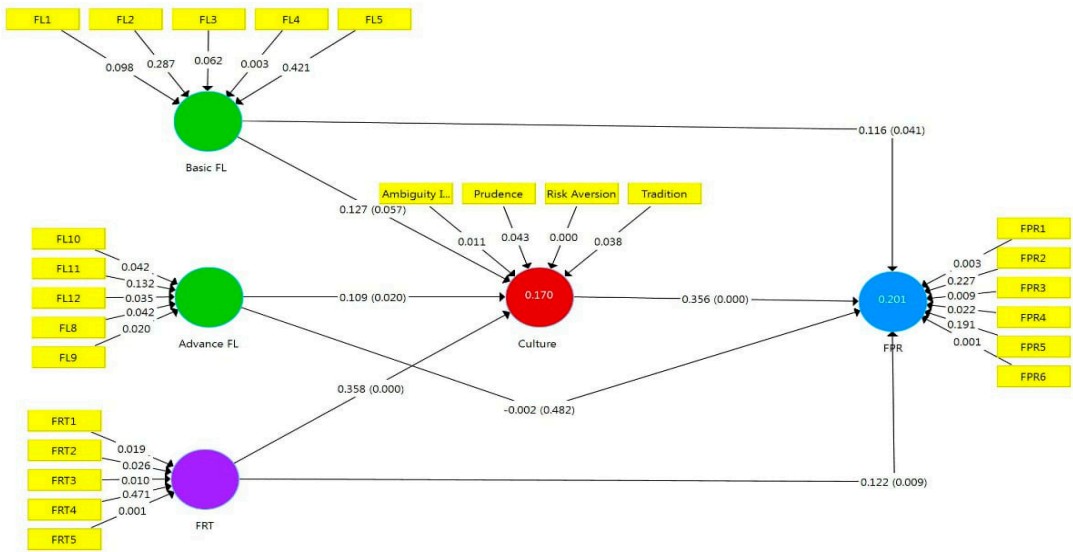

**Figure 2.** Hypothesis Testing: Bootstrapping Direct and Indirect Effect Results.

### 5.6. Multigroup Analysis (PLS-MGA)

Multigroup analysis (PLS-MGA) is a method that helps researchers examine pre-defined data sets by applying PLS-SEM to verify whether there are significant differences between group-specific parameter estimates (e.g., outer weights, outer loadings, and path coefficients) (Hair et al. 2017a). PLS-MGA was applied to evaluate the group differences in terms of path coefficients for the employment sector and gender. According to the multigroup and parametric tests in Table 7, only the causal relationship between financial risk tolerance and FPR ($H_3$) differed significantly across academics and administrators. This result indicates that the academic staff was more tolerant of risky financial investments in FPR than administrators (β academic > β administrators). However, Table 8 indicates that the study's findings hold for both genders (males and females), as the PLS-MGA test shows no significant gender impact on the study's empirical results.

**Table 7.** Results of PLS-MGA.

| Hypothesis | Path | Academic (330) | | Administrators (195) | | Parametric Test | | PLS-MGA | |
|---|---|---|---|---|---|---|---|---|---|
| | | Std. Beta | *p*-Value | Std. Beta | *p*-Value | Std. Beta | *p*-Value | Std. Beta | *p*-Value |
| $H_1$ | Basic FL -> FPR | −0.080 | 0.266 | 0.196 | 0.100 | −0.276 | 0.087 | −0.276 | 0.082 |
| $H_2$ | Advance FL -> FPR | 0.092 | 0.206 | 0.037 | 0.411 | 0.055 | 0.388 | 0.055 | 0.391 |
| $H_3$ | FRT -> FPR | 0.176 | 0.001 | −0.087 | 0.317 | 0.263 | 0.049 | 0.263 | 0.095 |
| $H_4$ | Culture -> FPR | 0.406 | 0.000 | 0.313 | 0.056 | 0.093 | 0.292 | 0.093 | 0.362 |

Source: Author Calculations.

**Table 8.** Results of PLS-MGA.

| Hypothesis | Path | Men (344) | | Women (181) | | Parametric Test | | PLS-MGA | |
|:---:|:---:|:---:|:---:|:---:|:---:|:---:|:---:|:---:|:---:|
| | | Std. Beta | *p*-Value | Std. Beta | *p*-Value | Std. Beta | *p*-Value | Std. Beta | *p*-Value |
| $H_1$ | Basic FL -> FPR | 0.097 | 0.102 | 0.182 | 0.186 | −0.085 | 0.319 | −0.085 | 0.279 |
| $H_2$ | Advance FL -> FPR | −0.076 | 0.252 | 0.197 | 0.146 | −0.273 | 0.093 | −0.273 | 0.127 |
| $H_3$ | FRT -> FPR | 0.106 | 0.051 | 0.117 | 0.238 | −0.011 | 0.470 | −0.011 | 0.439 |
| $H_4$ | Culture -> FPR | 0.449 | 0.000 | 0.245 | 0.069 | 0.204 | 0.076 | 0.204 | 0.103 |

Source: Author Calculations.

## 6. Discussion

This study aims to investigate the impact of cognitive, psychological, and external variables on financial retirement planning by applying the CWO model found by Hershey et al. (2012). The results of the current study are consistent with prior research, which shows that higher levels of the study's variables are associated with a greater tendency to plan and save for retirement. This study further provides insight into how culture acts as a mediator variable, given the limited tests conducted on culture in previous studies. Hence, the present study is unique in that various variables were examined simultaneously. These variables are often examined separately or combined with other variables, such as income, retirement goal clarity, or financial self-efficacy. Hence, the results of this study suggest that it would be beneficial to continue examining how various variables interact to shape individuals' preparation for retirement behaviors. By doing this, researchers can establish the relative impact of these predictive variables. Overall, while there are challenges associated with this type of research, it will contribute significantly to our understanding of the forces that motivate people to plan and save for retirement.

### 6.1. Consideration of Direct Effects

Return to the study's hypotheses, the SEM's analysis shows that basic financial literacy, financial risk tolerance, and culture have a significant and positive impact on the FPR, confirming the outcomes of preceding studies for basic financial literacy (Boisclair et al. 2017; Fornero and Monticone 2011; Lusardi and Mitchell 2008; Moure 2016; Ricci and Caratelli 2017; Sekita 2011) and financial risk tolerance (Jacobs-Lawson and Hershey 2005; Larson et al. 2016; Parker et al. 2012).

In the light of economic changes, recent changes to retirement systems in Saudi Arabia, based on the 2030 Vision and the financial difficulties individuals face (e.g., increase in the rate of unemployment, inflation, and living cost), basic knowledge of personal finance has assisted the respondents in preparing for their retirement. With respondents having basic financial literacy, it would imply that they potentially plan for retirement while they are still working, and that this has enabled them to effectively manage their financial resources prior to retirement to achieve post-retirement financial well-being. Referring to ICT, individuals were observed to be more likely to evaluate themselves so as to determine their strengths and weaknesses in the present as well as how to develop more ideal circumstances for their future (Boyatzis 2006).

Moreover, the result reveals that the respondents, whose answers were correct, may understand the meaning of the basic FL questions correctly (simple calculation, the effect of inflation, compound of interest rate, money illusion, time value of money) and that their responses are not expected to be guesswork. This means they have a positive attitude towards FPR, indicating their readiness for the period after they depart from the market. Therefore, basic FL is vital to retirement security.

While there was an expectation that there would be a relationship between the advanced FL and FPR, this relationship was not significant among respondents, as illustrated in Table 6. One reason that may have led to this result is the low level of financial literacy. Several types of research proved that a critical influence on retirement financial planning behavior is financial literacy (Sue Farrar et al. 2019; Klapper and Lusardi 2020). According to the previous studies, Saudi Arabia had a below-average level of financial literacy

(Alghamdi et al. 2021; Diaw 2017; Khan and Tayachi 2021; Mian 2014). Regarding the study's result, respondents' average financial literacy value was sufficiently literate for basic financial literacy (2.6 to 4) but less literate for advanced financial literacy (2.2 to 4)[1]. This is because the respondents of this study were drawn from a variety of educational backgrounds and not necessarily from business schools.

Another interpretation that may have led to this result is the role of excessive confidence among respondents. Individuals who have a high degree of confidence in their financial knowledge are more willing to estimate how much they should save for retirement (Van Rooij et al. 2012). The authors, however, are concerned that excess confidence in their financial literacy to manage complex retirement savings decisions may impede the ability of respondents to plan and calculate their retirement savings needs.

Regarding hypothesis three, the result indicated that FRT had influenced FPR behavior for public university employees, specifically among academics, as indicated in Table 7. This means that the degree of risk-taking that the respondents are ready to accept while jeopardizing current financial resources for future growth is above the average. This increase in FRT could be explained by the COVID-19 pandemic or by shifting FRT from financial decision-makers to individuals as a result of economic conditions. For example, higher-risk-tolerant academics tend to invest in higher-risk assets, such as equities and options, to achieve higher long-term returns. Hence, they had greater confidence in their financial knowledge, started bearing more financial risk attitudes, and were more likely to plan for risky financial investments and save for the future before leaving the workforce.

*6.2. Consideration of Indirect Effects*

The study's findings have shown the indirect influence of advanced financial literacy and financial risk tolerance on FPR through culture (Table 6). The estimation of the direct effect of advanced financial literacy on FPR was assessed as non-significant ($\beta = -0.002$, $p > 0.05$), although other studies have proved a significant relationship between them (Almenberg and Save-Soderbergh 2011; Baker et al. 2020; Brahmana et al. 2016; Van Rooij et al. 2011a). However, the standardized indirect effect (SIE) of advanced FL on FPR through culture was significant ($\beta = 0.039$, $p < 0.05$).

This outcome indicates uncertainty avoidance (e.g., high-risk aversion and ambiguity intolerance) and long-term orientation (e.g., tradition and prudence) as two cultural dimensions that are more likely to explain the relationship and variation between advanced FL and FRT regarding FPR. These two dimensions are responsible for the improvement of saving behaviors for those who are literate in personal finance background and take financial risks in achieving their goal of planning and saving for retirement. In other words, respecting traditional values, such as social awareness and morality, could lead those who know personal finance well and put themselves at financial risk to plan for long-term investment decisions and save efficiently, specifically after leaving the workforce.

## 7. Implication for Intervention

From a theoretical perspective, this study extends the implications of Intentional Change Theory (ICT), which may play an essential role in showing how individuals change their habits/behaviors from bad to desired ones (Boyatzis 2006) during their lifetime. The ICT may explain how employees intentionally start increasing their financial knowledge before reaching their retirement age. This study has extended the concept of this theory in view of the FPR context.

Regarding capacity variables, this research has provided valuable information for the literature on personal finances where basic financial literacy impacts FPR. It has shown the importance of cognitive factors in the theoretical and practical perspectives of public university employees when it comes to saving for retirement. Hence, there is no doubt that developing efficient courses in schools, universities, and workplaces to raise the level of financial knowledge and skills is necessary for FPR behavior. For example, the Ministry of Education should collaborate with the Public Pension Agency (PPA) and the General

Organization for Social Insurance (GOSI) to provide training programs for employees as a requirement to have a promotion in their job. Such training programs could be valuable for generals or professionals to share knowledge regarding FPR. Palací et al. (2017) highlighted that developing financial education by including courses to increase financial literacy and financial self-efficacy empowers individuals to understand the data they receive from newsletters, television, social media, and individuals.

In terms of external factors, this study has highlighted the role of culture. The variance of dimensions in the CWO model has explained the FPR behavior, and the culture as a mediator was confirmed as an antecedent of the FPR behavior. Obviously, culture is a critical variable that significantly affects the relationship between cognitive and psychological variables and FPR among public university employees. Encouraging positive work culture as the reflection of uncertainty avoidance and long-term orientation will improve the performance and sustainability of PPA policy as a significant part of Vision 2030. Therefore, increasing the value of culture is essential to inculcate retirement preparation behavior among workers.

## 8. Recommendations and Limitations

Apart from the contributions of the study, there is no research without limitations. Therefore, limitations and recommendations for ongoing research are discussed in the following section.

First, applying the CWO model to examine FPR behavior among individuals was only performed in Spain (Ghadwan et al. 2022). Other industries and emerging countries may not be identical in terms of the political, pension, and managerial systems, social life, and cultural characteristics (Palekar 2012). Such differences can significantly affect the outcomes of the CWO model. As a demonstration, the CWO model for this study provides empirical evidence of the impact of predictor variables on FPR behavior. To make existing results more generalizable, examining the CWO model in other developed countries (e.g., USA, UK, Australia) or in other emerging countries (e.g., Middle East countries, Malaysia, Brazil) will certainly make the results of the study more generalized. Future studies may also consider conducting a comparative study to discover employees' FPR behaviors in a variety of cultures.

Second, the Intentional Change Theory is a psychological theory (Boyatzis 2006). Applying ICT as a framework for FPR studies in the future will extend the use of the theory. Moreover, it may help to explain how workers change their FPR behaviors to desired ones. Third, this study only investigated the influence of financial literacy, financial risk tolerance, and culture on FPR behavior. Adding other cognitive (financial self-efficacy), psychological (retirement goal clarity), and external variables (financial resources) in the CWO model will help increase the coefficient of determination value for FPR and provide helpful insights into the relationship between these variables and FPR. Fourth, public university employees may have different behaviors and attitudes toward FPR compared to employees in private universities or other government sectors. Thus, examining those sectors for future research will also increase the generalization of the study results.

Finally, current research has used the quantitative method and the cross-sectional time horizon, which has proven to be appropriate for data analysis. Since this is not a longitudinal time horizon, the cross-sectional approach may not assist in identifying temporal and causal relationships between study variables at different periods. For example, the results revealed no relationship between advanced financial literacy and the FPR, even though it was documented in the literature. However, the longitudinal design could be applied to future research to increase generalization and would provide researchers with a better understanding of the relationships between variables that change over time.

## 9. Conclusions

To the best of the researcher's knowledge, this study could be the first to consider financial planning as a way to observe the retirement behaviors of public university em-

ployees in Saudi Arabia. This study also considered an array of capacities, psychological, and external characteristics to examine FPR. This outcome supports using the CWO model to examine FPR behavior from different disciplines. In this regard, the current study thus offers a practical and theoretical contribution to the current literature by presenting the most recent insights on FPR perception among higher education employees in Saudi Arabia.

**Author Contributions:** Conceptualization, A.G.; methodology, A.G.; software, A.G.; validation, A.G.; formal analysis, A.G.; investigation, A.G.; resources, A.G.; data curation, A.G.; writing—original draft preparation, A.G.; writing—review and editing, A.G.; visualization, A.G.; supervision, W.M.W.A. and M.H.H.; project administration, A.G. All authors have read and agreed to the published version of the manuscript.

**Funding:** This research received no external funding.

**Institutional Review Board Statement:** Not applicable.

**Informed Consent Statement:** Not applicable.

**Data Availability Statement:** Not applicable.

**Conflicts of Interest:** The authors declare no conflict of interest.

## Note

[1] Not literate (0–1.5); Less literate (1.5–2.5), sufficient literatre (2.5–3.5); Well literate (3.5–4).

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
