# Peer review of "Financial Planning for Retirement: The Mediating Role of Culture"

_risks, doi:10.3390/risks10050104_

Round 1
Reviewer 1 Report
The paper is well written and informative. However, there are some points that need to be addressed in more details:
- What threshold/technique have you used to identify outliers? Removing outliers could forgo some important information. Thus, perform the same analysis keeping all data in the model, and see whether the results hold, or how are they affected
- Describe in more details the CWO model, and Confirmatory Tetrad Analysis
- The authors should consider the efect of age on FPR. For instance, Jimenez et al. (2019) find that younger participants in Spain showed a greater level of FPR if they were characterized by a high level of education. Also, the analysis should be performed by considering separately men and women (see, e.g., Topa et al., 2018)
- „According to the United Nations Department of Economic and Social Affairs/Population Division (2017), around 13% of the world’s population will be 60 years and above by 2017” – this information is already obsolete and should be replaced with newest data if available, or reword
- There’s no need to mention statistical softwares employed in the analysis
References
Jiménez, I., Chiesa, R., Topa, G., 2019. Financial Planning for Retirement: Age-Related Differences Among Spanish Workers. Journal of Career Development 46, 550–566. https://doi.org/10.1177/0894845318802093
Topa, G., Segura, A., Pérez, S., 2018. Gender differences in retirement planning: A longitudinal study among Spanish Registered Nurses. Journal of Nursing Management 26, 587–596. https://doi.org/10.1111/jonm.12586
Reviewer 2 Report
The paper is interesting, concerning the relationship between Financial Literacy (FL), Financial Risk Tolerance (FRT), Culture, and Financial Planning for Retirement (FPR). It also investigates the mediating roles of culture on their relationship with FPR.
The subject is very interesting and the approach of structural equations modeling was, in my opinion, chosen well to try to propose a solution of the taken problem. Confirmatory Tetrad Analysis (CTA) allowed to uncovered variables contributing to financial planning determine for retirement practices and also to verify whether the conceptual model is formative or reflective.
In my opinion the aim, structure of the study, method of gathering and processing data are on a high level. So are the conclusions, supported with the discussion of the results with those obtained by other authors. References are up-to-date and well harmonized with the subject of the research.
I have a few technical and substantive suggestions:
1) After some subtitles and table titles (e.g. p. 5.4, table 2, 3 4 and others) there are dots (redundant). Please delete them. Also, there are no sources given under the tables (even if they are authors' own collaboration, it should be noticed).
2) Please explain, why in Table 5 some items for BFL and AFL are not significant, but they are kept?
3) In table 3, as it comes to cultural items scale, no Cronbach's alpha value is shown.
After explaining this I think the paper is of course worth publishing.
Reviewer 3 Report
The paper is interesting and devoted to urgent practical problems due to the aging of the population and the lack of financial literacy for appropriate financial planning after retirement. The method of SEM suits this type of the hypotheses testing, besides, it is aligned with the dataset and aim. However, some important details should be considered by authors before publishing their paper:
(1) Abstract should contain more specific information with data and peculiarities. Besides, it should be more logically presented according to the significance of the problem, novelty of the research, obtained results. I find it inappropriate to start from "The aim of this study...".
(2) Aim of the research should be more clear and it must reflect the results obviously. See notes for lines 228, 77, 8.
(3) Main drawbacks of the research are in the methodology description. First of all, the authors use samples structured by academic and non-academic staff. If differences are obtained, they should be mentioned. I believe the differences are significant due to the incomes, not only culture and financial awareness - these differences should be stressed; the second - the period of the research is omitted, only the period of Covid-19 is mentioned, but it is impossible to define the precise time horizon of the review; the third - it would be better if authors indicate the total number of public universities except for 29 participating on the research for in order to consider the scale of the study (see lines 234-235).
(4) grammar should be revised - see some notes in text, however, a proficient native speaker can find more.
(5) other notes see in the text.
Round 2
Reviewer 3 Report
Authors have considered comments in general, however, some important details are still omitted. I am surprised that some answers are provided only as responses for the reviewer but not included in the article.
First of all, the period of the research should be clearly stressed in the Methodology section, not only in the Abstract. This information is still missed.
The second. Authors deal with the information collected in two samples - see comment #5 in the Table with the Authors' response. I have achieved a detailed explanation of the logic of these samples formation. However, this explanation is available only as a response for the reviewer, it is not included in the paper. This gap should be filled in the text.
The style of responses is very surprising. Some of them are included in the text (like information about the total number of public universities etc.), but the authors find it necessary to explain them more obviously only for the reviewer in their comments.
And finally. Being interested in social policy problems for many years, I have a strong conviction that income is at least no less important for pension planning compared with culture and financial literacy. If you have no financial possibility to participate in some pension system, the culture and other personal attitudes would not create the appropriate financial background. Besides, the income level can have an indirect impact on financial risk tolerance. It is not an obligatory note, but I believe that your further research in the field would be more comprehensive with consideration of the income peculiarities. Now, this possibility could be added for further perspectives of research in the Recommendations and Limitations section.
